# Holographic covering and the fortuity of black holes

Chi-Ming Chang[a,b,c] and Ying-Hsuan Lin[d]

[a] *Yau Mathematical Sciences Center (YMSC), Tsinghua University, Beijing 100084, China*

[b] *School of Natural Sciences, Institute for Advanced Study, Princeton, NJ 08540, USA*

[c] *Beijing Institute of Mathematical Sciences and Applications (BIMSA)*
*Beijing 101408, China*

[d] *Jefferson Physical Laboratory, Harvard University, Cambridge, MA 02138, USA*

cmchang@tsinghua.edu.cn, yhlin@fas.harvard.edu

### Abstract

We propose a classification of BPS states in holographic CFTs into monotone and fortuitous, based on their behaviors in the large $N$ limit. Intuitively, monotone BPS states form infinite sequences with increasing rank $N$, while fortuitous ones exist within finite ranges of consecutive ranks. A precise definition is formulated using supercharge cohomology. We conjecture that under the AdS/CFT correspondence, monotone BPS states are dual to smooth horizonless geometries, and fortuitous ones are responsible for typical black hole microstates and give dominant contributions to the entropy. We present supporting evidence for our conjectures in the $\mathcal{N} = 4$ SYM and symmetric product orbifolds.

# 1 Introduction

In general relativity, there is a clear distinction between smooth horizonless geometries and black holes. Quantum mechanically, the former can be understood as coherent states of gravitons, while the nature of the latter is a subject of intense debate, epitomized by the information paradox. What is agreed upon is that any reasonable interpretation of black holes must explain the degrees of freedom constituting the Bekenstein-Hawking entropy. One hallmark success of string theory is to complete general relativity at high energies [1], to realize black holes as branes [2], and to account for the black hole entropy by the brane-localized fields [3]. Nevertheless, there remains a desire to understand black hole microstates in a framework where the low energy limit involves weakly coupled Einstein gravity.

To study the quantum nature of black holes, we adopt the framework of AdS/CFT [4], which postulates that all physical questions in quantum gravity in AdS can be addressed in the dual family of CFTs labeled by $N$. It provides a rigorous setting for studying gravitational dynamics using field theoretic techniques and for refining the Strominger-Vafa microstate counting beyond the aggregate properties of CFT states. In attempting to understand black hole microstates from a purely bulk perspective, one idea is that the space of smooth horizonless geometries with the same large-distance asymptotics as a black hole may be large enough to constitute the Bekenstein-Hawking entropy [5,6]. In reality, all the solutions constructed to date have not produced the desired entropy scaling. Is this a lamppost effect or is there a fundamental obstacle? The findings in this survey suggest the latter, that these geometries are not responsible for black holes, but rather encode highly nontrivial information about finite-$N$ quantum effects of graviton condensates.

To overcome the strong/weak nature of the AdS/CFT duality, supersymmetry has been indispensable, as it allows us to study various protected observables. In particular, the matching of superconformal indices has achieved tremendous success [7–11]. Somewhat less utilized are the "almost-protected" quantities or objects—those that jump at special high-codimension loci but remain constant elsewhere. One such object is the supercharge cohomology [12–14], which captures the complete spectrum of BPS operators. The recent revival of the black hole microstate program has been catalyzed by advances in this cohomological approach [15–22]. The supercharge cohomology contains much more refined information than the indices and allows us to analyze the relationship among operators at different $N$ that have the same formal representations (trace expressions in gauge theories and cycle shapes in symmetric product orbifolds).

What emerges from our analysis is a clean taxonomy of BPS operators into two classes, which we call *monotone* and *fortuitous*. Monotone BPS operators form infinite sequences with increasing rank $N$, while fortuitous ones exist within finite ranges of consecutive ranks and acquire anomalous dimensions away from those special ranks. As will be explained, the trace relations and their generalizations play a pivotal role in this classification. In the context of holography, we propose that monotone operators are dual to smooth horizonless geometries in the bulk, which is motivated by the following intuition: if a supersymmetric smooth horizonless geometry gives rise to a supersymmetric quantum state at $N = N_0$, then it is expected to remain a sensible supersymmetric quantum state at $N > N_0$, since the geometry should remain smooth and horizonless when the gravitational coupling $G_N$ becomes weaker as the mass and other conserved charges are kept constant.

We provide further evidence for this intuition by revisiting the dual geometries of half-BPS operators in $\mathcal{N} = 4$ SYM and those of chiral-chiral primary operators in symmetric product orbifolds. We demonstrate that the quantizations of their classical moduli spaces

correctly produce the Hilbert spaces of the corresponding monotone operators. Surprisingly, these geometries capture exact finite $N$ physics at low energies, even though their existence lies completely within the realm of classical gravity without the need for an ultraviolet completion such as string theory.

By contrast, supersymmetric geometries with horizons do not admit a regular weak coupling limit, in the sense that if the entropy is held fixed, then the horizon shrinks as $G_N \to 0$. This resonates with the fact that fortuitous operators do not have natural lifts to large $N$. By comparing the dimensions of the bulk graviton Fock space with the superconformal index, we know that monotone operators are greatly outnumbered by fortuitous ones. In particular, a typical black hole microstate is fortuitous. This perspective may open a new path toward constructing black hole microstates.

# 2 Holographic covering and BPS taxonomy

## 2.1 What is a holographic CFT?

When speaking of a holographic CFT, the minimal defining data consists of a sequence of CFTs with increasing rank $N$ and increasing conformal central charge $C_N$. From this data alone, it is unclear what sequences of observables we should consider in taking the large $N$ limit to have nice holographic duals, and whether the $e^{N^\#}$ terms in the large $N$ expansion can be unambiguously extracted to compare with bulk subleading saddles and brane/instanton contributions. In practice, there are usually three approaches to taking large $N$: (1) studying partition functions without operator insertions; (2) leaving the precise sequence of observables unspecified and resorting to genericity arguments; (3) working with a concrete Lagrangian where operators have natural representations, and considering sequences of operators of the same form. While (1) and (2) have taught us fruitful lessons, it is (3) that is more in line with our belief that AdS/CFT is an *exact* duality and allows us to do more than matching aggregate quantities. A major goal of this paper is to attempt to formalize the idea of a "natural sequence of operators".

A potential way to unify across $N$ is to define a universal covering "CFT" in terms of a Hilbert space $\widetilde{\mathcal{H}}(N)$ of local operators and an operator product algebra $\widetilde{A}(N)$ depending on a *continuous* parameter $N$. At positive integer values of $N$, the algebra $\widetilde{A}(N)$ admits ideals $I_N$ by which one could quotient to define the Hilbert spaces $\mathcal{H}_N$ and operator product algebras $A_N$ of finite $N$ theories. While we do not pursue the ambitious goal of formulating $\widetilde{\mathcal{H}}(N)$ and $\widetilde{A}(N)$, it is reassuring to note that the emblematic holographic CFT—the $\mathcal{N} = 4$ super-Yang-Mills with SU($N$) gauge group—appears to have this structure at weak coupling, where $\widetilde{\mathcal{H}}$ is spanned by formal multi-trace operators, $\widetilde{A}$ is the operator product of multi-traces

in the 't Hooft $(1/N, \lambda)$ perturbation theory, and $I_N$ are the trace relations.

In the following, we focus on the Hilbert spaces and discuss how BPS states across different $N$ are related. To simplify our analysis further, we forget about the norm and regard $\mathcal{H}_N$ and $\widetilde{\mathcal{H}}$ as vector spaces. It turns out that this is already sufficient to shed light on new aspects of BPS states.

## 2.2 Holographic covering hypothesis

We are interested in the relations among the Hilbert spaces $\mathcal{H}_N$ at different ranks $N$ in the large $N$ limit. There are natural identifications across $N$ for distinguished operators like the stress tensor and symmetry currents. Furthermore, a basic check of any instance of holography is to identify the duals of perturbative excitations in AdS as sequences of operators with increasing $N$, and this identification gives us a natural organization of light operators across $N$. Our first goal is to formalize the organization of light operators across $N$ for a large class of holography CFTs. For heavy operators with conformal dimensions that grow with $N$, it is unclear whether there are natural sequences of operators relevant to holography, and this question is beyond the scope of this survey.[1]

**Definition 1** (Covering). *Given a sequence of vector spaces $\mathcal{H}_N$, $N = 1, 2, \ldots$, a covering consists of a covering vector space $\widetilde{\mathcal{H}}$ and a sequence of subspaces $I_N$ such that[2]*

$$\mathcal{H}_N \simeq \widetilde{\mathcal{H}}/I_N \,, \tag{2.1}$$

*or equivalently, there is a short exact sequence*

$$0 \longrightarrow I_N \overset{i_N}{\hookrightarrow} \widetilde{\mathcal{H}} \overset{\pi_N}{\longrightarrow} \mathcal{H}_N \longrightarrow 0. \tag{2.2}$$

**Definition 2** (Strictness). *A covering is strict if the following two conditions are satisfied:*

*(a) The subspaces $I_N$ at adjacent ranks are related by strict inclusions*

$$I_N \supsetneq I_{N+1} \,. \tag{2.3}$$

*(b) For any vector $v \in \widetilde{\mathcal{H}}$, there exists an integer $N'$ such that $v \notin I_{N'}$.*

---

[1]It has been argued that the lack of such sequences for black hole microstates is crucial for the phenomenon of chaos [23].

[2]The vector spaces $\mathcal{H}_N$ will be taken to be the physical Hilbert spaces. If we want to further recover the norm from the quotient (2.1), we need to define a norm for the covering vector space $\widetilde{\mathcal{H}}$ (more precisely, a family of norms with continuous dependence on $N$). However, because the norm of $\mathcal{H}_N$ will play little role in this section, we do not introduce a norm for $\widetilde{\mathcal{H}}$ here.

**Definition 3** (Covering symmetry). *Given a compact group $G$ and its representations $r_N$ on $\mathcal{H}_N$ for every $N$, a covering symmetry is a representation $\widetilde{r}$ on $\widetilde{\mathcal{H}}$ that is compatible with the short exact sequence* (2.2), *meaning that for every $g \in G$, $\widetilde{r}(g)I_N \subset I_N$ and $\pi_N(r(g)v) = r_N(g)\pi_N(v)$ for all $v \in \widetilde{\mathcal{H}}$.*

There are many examples of holographic CFTs admitting coverings, including $\mathrm{SU}(N)$ gauge theories with adjoint matter (in particular, $\mathcal{N} = 4$ SYM) and symmetric product orbifolds.[3] We will study these examples in Section 4. While the strict covering of a sequence of $\mathcal{H}_N$ may not be unique, Definition 2b is such that the large $N$ limit depends on the choice of strict covering only in a mild way.

Since vector spaces of the same dimension are all isomorphic, the above definitions are somewhat vacuous if we identify $\mathcal{H}_N$ with the infinite-dimensional space of all operators in a holographic CFT. To remedy this, we introduce supersymmetry and consider the space $\mathcal{H}_N^{\mathrm{BPS}}$ of BPS operators, since this space decomposes into a direct sum of *finite*-dimensional subspaces upon symmetry refinement. By now, various twisted formalisms endow BPS sectors with algebras, and it may be possible to formalize their holographic coverings at the algebraic level. However, as a first step, we will be content with holographic coverings at the vector space level, and derive ramifications on the dimensionality of vector spaces.

Consider a holographic SCFT with vector spaces $\mathcal{H}_N$ of local operators. For each $N$, let $Q_N$ be a supercharge whose Hermitian (BPZ) conjugate is a conformal supercharge $S_N = Q_N^\dagger$. The BPS subspace $\mathcal{H}_N^{\mathrm{BPS}} \subset \mathcal{H}_N$ is the intersection of the kernels of $Q_N$ and $Q_N^\dagger$. By standard arguments of Hodge theory [26,13], $\mathcal{H}_N^{\mathrm{BPS}}$ is isomorphic to the cohomology of the supercharge $Q_N$, i.e.

$$h : H_{Q_N}^*(\mathcal{H}_N) \xrightarrow{\simeq} \mathcal{H}_N^{\mathrm{BPS}} \,. \tag{2.4}$$

This isomorphism provides a powerful tool for studying BPS operators because it does not require the knowledge of $Q_N^\dagger$ to define and compute the $Q_N$-cohomology. However, computing $H_{Q_N}^*(\mathcal{H}_N)$ at generic finite coupling is still a formidable task. Fortunately, it has been conjectured with ample evidence that the BPS spectra are constant along conformal manifolds away from the free points [13,14,18,27–30]. Furthermore, this conjecture has been partially proven assuming the existence of a basis of local operators such that the action of the supercharge $Q_N$ in this basis is not renormalized quantum mechanically [15].[4] Under this premise, one could compute $H_{Q_N}^*(\mathcal{H}_N)$ at weak coupling.

---

[3]An example of a holographic CFT that is not known to admit a covering is the $\mathcal{N} = 4$ SYM with $\mathrm{SO}(2N)$ gauge group, due to the presence of Pfaffian operators [24,25]. However, we can focus on a subsector of the theory that admits a (strict) covering and the discussions in this section apply.

[4]Counter-examples can be found in 4d $\mathcal{N} = 1$ non-conformal SQFTs, but no conformal counter-examples exist to the authors' knowledge. We thank Jingxiang Wu and Kasia Budzik for discussions on this point.

Suppose a holographic SCFT $(\mathcal{H}_N, Q_N)$ has a covering $(\widetilde{\mathcal{H}}, \widetilde{Q}, \pi_N)$ where the supercharge commutes with the quotient map $\pi_N$, i.e. $\pi_N \widetilde{Q} = Q_N \pi_N$.[5] The supersymmetry action on the short exact sequence (2.2) induces a long exact sequence [14]

$$\cdots \longrightarrow H^n(I_N) \xrightarrow{i_{N*}} H^n(\widetilde{\mathcal{H}}) \xrightarrow{\pi_{N*}} H^n(\mathcal{H}_N) \xrightarrow{f} H^{n+1}(I_N) \longrightarrow \cdots , \qquad (2.5)$$

where $H^* := H^*_{\widetilde{Q}}$ or $H^*_{Q_N}$, $n$ is a grading for the $Q$-cohomology with $n(Q) = 1$. The structure of the $Q$-cohomology motivates the following taxonomy of BPS states.

**Definition 4** (Monotone and fortuitous). *Suppose there exists a covering $(\widetilde{\mathcal{H}}, \widetilde{Q}, \pi_N)$ of a sequence $(\mathcal{H}_N, Q_N)$ of SCFTs. At a given $N$, a vector in $\mathcal{H}_N^{\mathrm{BPS}}$ is a monotone BPS state if it is inside $h \circ \operatorname{im} \pi_*$, and the subspace of all such states is denoted by $\mathcal{H}_N^{\mathrm{mon}}$. A vector in $\mathcal{H}_N^{\mathrm{BPS}}$ a fortuitous BPS state if it lies in the orthogonal complement of $\mathcal{H}_N^{\mathrm{mon}}$, and this subspace is denoted by $\mathcal{H}_N^{\mathrm{fts}}$. In summary, we have*

$$\begin{aligned}
\mathcal{H}_N^{\mathrm{mon}} &\simeq \operatorname{im} \pi_{N*} \simeq H^*(\widetilde{\mathcal{H}})/\operatorname{im} i_{N*} , \\
\mathcal{H}_N^{\mathrm{fts}} &\simeq \operatorname{im} f \simeq H^*(\mathcal{H}_N)/\operatorname{im} \pi_{N*} .
\end{aligned} \qquad (2.6)$$

The choice of names "monotone" and "fortuitous" is motivated by the qualitative pattern when lifting an operator successively from $\mathcal{H}_N$ to $\mathcal{H}_{N+1}, \mathcal{H}_{N+2}, \ldots$ using the long exact sequence (2.5), and is also related to the behavior of the anomalous dimension as $N$ is continuously varied. This will be explained in the next subsection.

Let us rephrase Definition 4 and describe the monotone and fortuitous BPS operators in terms of $I_N$, which are the generalizations of trace relations in matrix theories. Given a monotone BPS operator $\mathcal{O} \in \mathcal{H}_N^{\mathrm{mon}}$, we can find a lift $\widetilde{\mathcal{O}} \in \widetilde{\mathcal{H}}$ of the operator $\mathcal{O}$, i.e. $\pi(\widetilde{\mathcal{O}}) = \mathcal{O}$, such that $\widetilde{\mathcal{O}}$ represents a nontrivial cohomology class in $H^*(\widetilde{\mathcal{H}})$; in other words,

$$\text{Monotone:} \quad \exists\, \widetilde{Q} \ \text{such that} \ Q\widetilde{\mathcal{O}} = 0. \qquad (2.7)$$

Next, given a fortuitous BPS operator $\mathcal{O} \in \mathcal{H}_N^{\mathrm{fts}}$. Unlike the previous case, for all lifts $\widetilde{\mathcal{O}} \in \widetilde{\mathcal{H}}$ of $\mathcal{O}$, the operator $\widetilde{\mathcal{O}}$ is not $Q$-closed but equals a non-zero element in $I_N$, i.e.

$$\text{Fortuitous:} \quad \forall\, \widetilde{\mathcal{O}}, \ Q\widetilde{\mathcal{O}} \neq 0 \ \text{and} \ Q\widetilde{\mathcal{O}} \in I_N . \qquad (2.8)$$

Since $\widetilde{\mathcal{O}} \notin I_N$, $Q\widetilde{\mathcal{O}}$ represents a nontrivial cohomology class in $H^*(I_N)$.

## 2.3 Monotone versus fortuitous sequences

From now on we assume a strict holographic covering. Given an operator $\mathcal{O}_N \in \mathcal{H}_N^{\mathrm{BPS}}$ and a (non-unique) lift $\widetilde{\mathcal{O}} \in \widetilde{\mathcal{H}}$, we get a (non-unique) sequence of operators

$$\mathcal{O}_N, \ \pi_{N+1}(\widetilde{\mathcal{O}}), \ \pi_{N+2}(\widetilde{\mathcal{O}}), \ \cdots . \qquad (2.9)$$

---

[5] Since we did not introduce a norm for $\widetilde{\mathcal{H}}$, we cannot define a $Q^\dagger$-action on $\widetilde{\mathcal{H}}$.

1. If $\mathcal{O}_N$ is a monotone BPS operator, then the operators $\pi_{N+i}(\widetilde{\mathcal{O}})$ for $i > 0$ represent nontrivial cohomology classes in $H^*(\mathcal{H}_{N+i})$. This is because if $\pi_{N+i}(\widetilde{\mathcal{O}})$ is $Q$-exact, then by the exactness of the sequence (2.5), there exists $\widetilde{\mathcal{O}}' \in \widetilde{\mathcal{H}}$ such that $\widetilde{\mathcal{O}} + Q\widetilde{\mathcal{O}}' \in I_{N+i} \subset I_N$, and hence $\pi_N(\widetilde{\mathcal{O}})$ is also $Q$-exact, contradicting our assumption. Therefore, we have an *infinite* sequence of monotone BPS operators

$$\mathcal{O}_N, \, \mathcal{O}_{N+1}, \, \mathcal{O}_{N+2}, \, \cdots , \tag{2.10}$$

where $\mathcal{O}_{N+i} = h \circ [\pi_{N+i}(\widetilde{\mathcal{O}})]$ with $h$ the Hodge isomorphism (2.4), and the bracket $[\mathcal{O}]$ denotes the $Q$-cohomology class represented by $\mathcal{O}$.

2. Suppose $\mathcal{O}_N$ is a fortuitous BPS operator. If $Q\widetilde{\mathcal{O}} \in I_{N+i}$, then $\pi_{N+i}(\widetilde{\mathcal{O}})$ represents a nontrivial cohomology class in $H^*(\mathcal{H}_{N+i})$ by the preceding argument, but if $Q\widetilde{\mathcal{O}} \notin I_{N+i}$, then $\pi_{N+i}(\widetilde{\mathcal{O}})$ is not $Q$-closed. By strictness, there exists a positive integer $N_{\max}$ such that $Q\widetilde{\mathcal{O}} \in I_{N+i}$ for $N + i \leq N_{\max}$, and $Q\widetilde{\mathcal{O}} \notin I_{N+i}$ for $N + i > N_{\max}$. Hence, we only have a *finite* sequence of fortuitous BPS operators

$$\mathcal{O}_N, \, \mathcal{O}_{N+1}, \, \mathcal{O}_{N+2}, \, \cdots , \mathcal{O}_{N_{\max}} . \tag{2.11}$$

We can extend this sequence by appending the operators $\pi_{N+i}(\widetilde{\mathcal{O}})$ for $N + i > N_{\max}$, but these operators are not BPS.

If we introduce the structure of $\widetilde{Q}^\dagger$ on $\widetilde{\mathcal{H}}$, then the holographic covering allows one to consider the anomalous dimension of $\widetilde{\mathcal{O}}$ as $N$ is continuously varied. In the bulk, if we ignore flux quantization, then this corresponds to the adiabatic process of varying $G_N$ while fixing the charges. We expect that a monotone BPS state has identically zero anomalous dimension, whereas a fortuitous BPS state has generically nonzero anomalous dimension except in a finite range $N$, $N + 1$, ..., $N_{\max}$. In Figure 1, we depict the typical behaviors of the lifts of monotone and fortuitous BPS operators. In [20], these behaviors were explicitly seen by analyzing the one-loop anomalous dimensions of operators in the $\mathcal{N} = 4$ SYM with $SU(N)$ gauge group. The discussions above explain the names "monotone" and "fortuitous".

# 3    Lessons and conjectures

We now present several consequences and conjectures for strict holographic coverings of SCFTs.

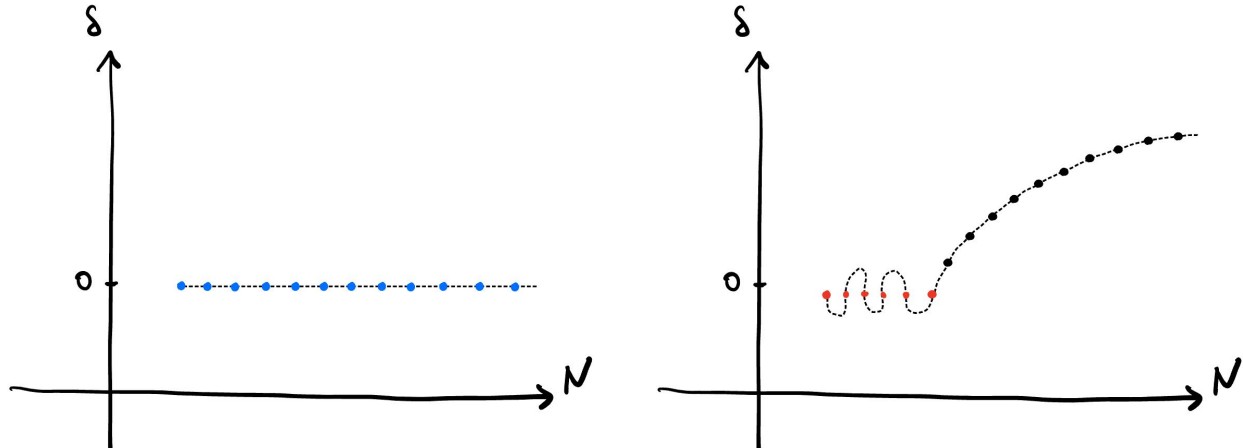

Figure 1: Typical behaviors of sequences from lifting monotone and fortuitous BPS operators to larger $N$, where $\delta$ is the anomalous dimension. The dashed lines indicate their anomalous dimensions continued to non-integer $N$ in perturbation theory.

## 3.1 Perturbative bulk Hilbert space

The sequence of monotone BPS operators (2.10) "stabilizes" in the following sense. Let us denote the global symmetry charges collectively by $q$, and let $[\mathcal{O}_N] \in \operatorname{im} \pi_*|_q$ be the cohomology class represented by the monotone BPS operator $\mathcal{O}_N$. It could be lifted to a cohomology class $[\widetilde{\mathcal{O}}] \in H^n(\widetilde{\mathcal{H}})_q$, with ambiguities residing in the space $\ker \pi_*|_q \simeq \operatorname{im} i_*|_q$ by the long exact sequence (2.5). Now, we argue that the ambiguity $\operatorname{im} i_*|_q$ is trivial for large enough $N$. By a similar argument as for (2.4), the cohomology $H^*(I_N)_q$ is isomorphic to the subspace of $I_{N,q}$ that is annihilated by both $Q_N$ and $Q_N^\dagger$. By strictness, such a subspace is expected to be trivial for large enough $N$; hence, $\operatorname{im} i_*|_q$ is also trivial, and the monotone BPS operator $\mathcal{O}_N$ has a unique lift $[\widetilde{\mathcal{O}}] \in H^*(\widetilde{\mathcal{H}})_q$.

Under the AdS/CFT correspondence, a sequence of local operators with order one conformal dimensions maps in the large $N$ limit to a perturbative state in the AdS background. Hence, we have an injection from $H^*(\widetilde{\mathcal{H}})$ into the Hilbert space $\mathcal{H}_{\mathrm{BPS}}^{\mathrm{pert}}$ of perturbative BPS states, which is given by the Fock space of BPS particles in AdS.[6] Potentially, there could be particles or bound states of particles that saturate the BPS bound at infinite $N$ and are yet non-BPS at large finite $N$, but we are unaware of such examples. It is plausible that with enough supersymmetry, there are non-renormalization theorems that prevent $1/N$ corrections in Witten diagram computations. We thereby conjecture that this map is also surjective.

**Conjecture 1.** *The cohomology $H^*(\widetilde{\mathcal{H}})$ is isomorphic to $\mathcal{H}_{\mathrm{BPS}}^{\mathrm{pert}}$, the Fock space of BPS par-*

---

[6]We only consider particles satisfying the BPS condition for the same supercharge.

*ticles in AdS, i.e.*

$$\mathcal{H}_{\mathrm{BPS}}^{\mathrm{pert}} \simeq H^*(\widetilde{\mathcal{H}}) \,. \tag{3.1}$$

## 3.2  Finite $N$ quotient from quantizing supergravity solutions

One could consider more general sequences of monotone BPS operators, in particular, sequences with simultaneously increasing charge $q$ and rank $N$. We could lift this sequence (possibly with ambiguities) to a sequence of cohomology classes in $H^*(\widetilde{\mathcal{H}})$, which is holographically dual to a sequence of states in the perturbative bulk BPS Hilbert space by Conjecture 1. Therefore, such sequences could be interpreted as coherent states or condensates of gravitons (and other light particles) with large back reactions. In other words, from the classical bulk perspective, they are non-linear completions of linear solutions to the supergravity equations that are continuously connected to empty AdS. Furthermore, we expect these solutions to be horizonless since the existence of a horizon and a curvature singularity means that the curvature operator acting on the bulk quantum state diverges. If we believe that all smooth horizonless geometries are coherent states of this sort, then we arrive at the following conjecture.

**Conjecture 2.** *Under the AdS/CFT correspondence, the space $\mathcal{H}_N^{\mathrm{mon}}$ of monotone BPS operators is given by a quantization of the classical moduli space of supersymmetric, smooth and horizonless solutions to the supergravity equations with all fluxes held fixed.*

The rank $N$ is related to the fluxes of the solutions, but the precise relation is specific to each case. Since a classical system can have multiple consistent quantizations, some choice of quantization may be necessary in the above conjecture. However, in the next section, we will find that for certain higher-BPS solutions, Conjecture 2 is realized through standard quantization procedures, e.g. geometric quantization and deformation quantization. From Conjecture 1, 2 and the isomorphisms in (2.6), we see that the Hilbert space of quantized supersymmetric smooth horizonless geometries is a quotient of the Hilbert space of perturbative supersymmetric excitations.

## 3.3  On the fortuity of black holes

From the long exact sequence (2.5), it is clear that the number of monotone BPS operators is bounded by the number of cohomology classes in $H^n(\widetilde{\mathcal{H}})$. Let us consider the subspaces of operators or $Q$-cohomology classes that have fixed global symmetry charges (including R-charges and angular momenta), which we collectively denote by $q$. With this refinement, the spaces $\mathcal{H}_{N,q}^{\mathrm{mon}}$ and $\mathcal{H}_{N,q}^{\mathrm{fts}}$ become finite-dimensional, and we have the following theorem.

**Theorem 1.** *In the symmetry-refined long exact sequence* (2.5), *there exist the following bounds:*

$$
\begin{aligned}
\dim(\mathcal{H}_{N,q}^{\text{mon}}) &= \dim(\operatorname{im}\pi_*|_q) \leq \dim(H^*(\widetilde{\mathcal{H}})_q)\,, \\
\dim(\mathcal{H}_{N,q}^{\text{fts}}) &= \dim(\operatorname{im}f|_q) \leq \dim(H^*(I_N)_q)\,.
\end{aligned}
\tag{3.2}
$$

To make contact with black hole physics, consider a large $N$ limit where the charge $q$ is scaled as that of a black hole $q_{\text{BH}} \sim N^{\frac{d}{2}}$ with $d$ the spacetime dimension of the holographic CFT. Suppose the bulk dual of the CFT admits a supergravity limit as $\text{AdS}_{d+1} \times M_p$ (with potential warping), then the dimension of the perturbative BPS Hilbert space $\mathcal{H}_{\text{BPS}}^{\text{pert}}$ at large charge $q$ should be bounded by

$$
\log\left[\dim(\mathcal{H}_{\text{BPS},q}^{\text{pert}})\right] \lesssim S_{\text{pert}} \sim q^{\frac{d+p}{d+p+1}}\,,
\tag{3.3}
$$

where we used the entropy scaling of a gas of free particles in $d + p + 1$ dimensions. Now, let us set $q \sim q_{\text{BH}} \sim N^{\frac{d}{2}}$. Using the bound (3.2) and Conjecture 1, we find

$$
\dim(\mathcal{H}_{N,q_{\text{BH}}}^{\text{mon}}) \leq \dim(H^*(\widetilde{\mathcal{H}})_{q_{\text{BH}}}) = \dim(\mathcal{H}_{\text{BPS},q_{\text{BH}}}^{\text{pert}}) \lesssim \exp(N^{\frac{d(d+p)}{2(d+p+1)}})\,.
\tag{3.4}
$$

On the other hand, if the bulk also admits BPS black hole solutions, then the dimension of the space $\mathcal{H}_{N,q_{\text{BH}}}^{\text{BPS}}$ should grow as

$$
\dim(\mathcal{H}_{N,q_{\text{BH}}}^{\text{BPS}}) = e^{S_{\text{BH}}} \sim \exp(N^{\frac{d}{2}})\,.
\tag{3.5}
$$

Comparing (3.4) and (3.5), we arrive at the following conjecture.

**Conjecture 3.** *In the large $N$ limit, the number of fortuitous BPS operators is exponentially larger than that of monotone BPS operators.*

Conjecture 3 further lends credence to Conjecture 2: since the number of monotone BPS states with charge $q \sim q_{\text{BH}} \sim N^{\frac{d}{2}}$ is exponentially smaller than the number of all BPS states, the number of operators in $\mathcal{H}_N^{\text{mon}}$ alone cannot support any horizon.

# 4 Case studies

In this section, we study two large classes of holographic CFTs with strict holographic coverings, $\text{SU}(N)$ gauge theories with adjoint matter and symmetric product orbifolds, and gather concrete evidence for the conjectures.

## 4.1  $\mathcal{N} = 4$ SYM and the deformation quantization of LLM geometries

Consider a weakly coupled $SU(N)$ gauge theory with adjoint matter. The Hilbert space of local operators is spanned by the multitraces of fundamental fields in the adjoint representation, modulo trace relations. It is then natural to define the covering Hilbert space $\widetilde{\mathcal{H}}$ as the space of formal multitrace expressions without imposing the trace relations, and $I_N$ as the space of trace relations at rank $N$. It is easy to see that this covering is strict: (1) the trace relations at a given rank are also trace relations at lower ranks, while decreasing the rank always introduces new trace relations, so we have the inclusion (2.3), and (2) given a trace expression $\mathcal{O} \in \widetilde{\mathcal{H}}$ where the longest trace is of a product of $L$ matrices, then $\mathcal{O}$ cannot be inside the space $I_N$ with $N > L$. We have thus constructed a strict holographic covering.

Now, let us focus on the $\mathcal{N} = 4$ SYM. The $Q$-cohomology was first studied in [12–14], and more recently in [15–22]. The cohomology $H^*(\widetilde{\mathcal{H}})$ is generated by the single-trace cohomology classes, whose representatives have been explicitly found and the spectrum exactly matched with that of perturbative single-graviton excitations on $\text{AdS}_5 \times \text{S}^5$ [14]. This gives strong evidence for Conjecture 1. The cohomology classes in $\operatorname{im} \pi_*$, corresponding to the monotone BPS operators, are given by imposing trace relations on the cohomology classes in $H^*(\widetilde{\mathcal{H}})$.

Unlike the monotone BPS operators, there is no known systematic way to find all the fortuitous BPS operators. The first examples of fortuitous BPS states were discovered by the present authors in [15] in the SU(2) gauge theory, by explicitly computing the cohomology classes of $H^n(\mathcal{H}_2)$ to high charges. Simple representatives for this cohomology class as well as additional infinite families were later found in [16, 17, 22].

To make contact with holography, consider the number of BPS states in the large $N$ limit with the charges (collectively denoted by $q$) scaled as $q \sim N^2$. The number of monotone BPS states in the large $q$ limit is bounded by [14]

$$\dim(\mathcal{H}_{\text{mon},q}) < \dim H^n(\widetilde{\mathcal{H}})_q \sim e^{q^{\frac{5}{6}}}, \tag{4.1}$$

which implies the bound

$$(\# \text{ monotone BPS}) \lesssim e^{N^{\frac{5}{3}}}. \tag{4.2}$$

On the other hand, by studies of the large $N$ limit of the superconformal index [9, 10], we know that the number of all BPS states should scale as

$$(\# \text{ all BPS}) \sim e^{N^2}. \tag{4.3}$$

There are exponentially more fortuitous BPS operators than monotone ones in the $\mathcal{N} = 4$ SYM, and Conjecture 3 holds.

Next, we present evidence for Conjecture 2. Monotone BPS operators form a very large class of BPS operators. In particular, it plausibly exhausts all the BPS operators in the three different $\frac{1}{8}$-BPS subsectors: $\mathfrak{su}(2|3)$, $\mathfrak{su}(1,1|2)$, and $\mathfrak{su}(1,2|2)$ [13, 21]. Let us focus on the BPS operators in the $\mathfrak{su}(2|3)$ subsector, which are the chiral primaries with respect to $\mathcal{N} = 1$ supersymmetry. The partition function of these $\frac{1}{8}$-BPS operators is [12]

$$Z_N^{\frac{1}{8}\text{-BPS}} = \prod_{m,n,r=0}^{\infty} \frac{(1 - pzq_1^{2m+1}q_2^{2n+1}q_3^{2r+1})(1 - pz^{-1}q_1^{2m+1}q_2^{2n+1}q_3^{2r+1})}{(1 - pq_1^{2m}q_2^{2n}q_3^{2r})(1 - pq_1^{2m+2}q_2^{2n+2}q_3^{2r+2})}\bigg|_{p^N}, \tag{4.4}$$

where $z$ and $q_i$ are the fugacities of the $\mathrm{SU}(2)_R \subset \mathrm{SO}(4)$ rotation symmetry and the $\mathrm{SU}(3) \subset \mathrm{SU}(4)$ R-symmetry. One could further specialize to smaller subsectors with more supersymmetry, including the $\frac{3}{16}$-BPS $\mathfrak{su}(1|2)$ sector, the $\frac{1}{4}$-BPS $\mathfrak{su}(2)$ sector, and the $\frac{1}{2}$-BPS $\mathfrak{u}(1)$. The corresponding partition functions are

$$Z_N^{\frac{3}{16}\text{-BPS}} = \prod_{m,n=0}^{\infty} \frac{1 - pzq_1^{2m+1}q_2^{2n+1}}{1 - pq_1^{2m}q_2^{2n}}\bigg|_{p^N},$$

$$Z_N^{\frac{1}{4}\text{-BPS}} = \prod_{m,n=0}^{\infty} \frac{1}{1 - pq_1^{2m}q_2^{2n}}\bigg|_{p^N}, \tag{4.5}$$

$$Z_N^{\frac{1}{2}\text{-BPS}} = \prod_{n=1}^{N} \frac{1}{1 - q^{2n}}.$$

Conjecture 2 asserts that the Hilbert spaces $\mathcal{H}_N^{\frac{1}{8}\text{-BPS}}$, $\mathcal{H}_N^{\frac{3}{16}\text{-BPS}}$, $\mathcal{H}_N^{\frac{1}{4}\text{-BPS}}$, $\mathcal{H}_N^{\frac{1}{2}\text{-BPS}}$ can be recovered from the bulk by quantizing the classical moduli spaces of supergravity solutions. The supergravity solutions preserving 16 supercharges ($\frac{1}{2}$-BPS) are the famous Lin-Lunin-Maldacena (LLM) geometries [31], and the $\frac{1}{4}$ and $\frac{1}{8}$-BPS generalizations of the LLM geometries were found in [32–36]. The quantization of the classical moduli space of the LLM geometries was carried out in [37, 38], which recovered the Hilbert space $\mathcal{H}_{N=\infty}^{\frac{1}{2}\text{-BPS}} \simeq \widetilde{\mathcal{H}}^{\frac{1}{2}\text{-BPS}}$. Below, we improve upon their quantization to recover the Hilbert space $\mathcal{H}_N^{\frac{1}{2}\text{-BPS}}$ at *finite $N$*. We leave the quantization of less supersymmetric geometries for future work.

The Lin-Lunin-Maldacena (LLM) geometries are solutions to Type IIB supergravity that are asymptotically $\mathrm{AdS}_5 \times \mathrm{S}^5$ and preserve 16 supercharges [31]. They are parametrized by a $\mathbb{Z}_2$-valued function $u(x_1, x_2)$ (i.e. $u = 0, 1$) on a two-plane $\mathbb{R}^2$. We will refer to the $u = 1$ and $u = 0$ regions as black and white. When fixing the five-form flux on the asymptotic $\mathrm{S}^5$ to $N$, the area of the black region is fixed as

$$\sqrt{\hbar_{10}}N = \frac{1}{\pi\kappa} \int_{\mathrm{S}^5} F_5 = \frac{1}{2\pi\kappa} \int u(x)\, d^2x = \frac{1}{2\pi\kappa}A, \tag{4.6}$$

where $\kappa := \kappa_{10}/4\pi^{\frac{5}{2}}$, and $\hbar_{10}$, $\kappa_{10}$ are the ten-dimensional Planck and gravitational constants. In [31,37,38], the Planck constant was set to unity, i.e. $\hbar_{10} = 1$, but here we choose to restore it for clarity. The flux quantization gives $N \in \mathbb{Z}_{\geq 0}$, and we will see later that this is consistent with the quantization of the classical moduli space. The energy (ADM mass) of the LLM geometry over the $\mathrm{AdS}_5 \times \mathrm{S}^5$ is

$$H = \frac{1}{4\pi\kappa^2} \int (x_1^2 + x_2^2) u(x)\, d^2x - \frac{1}{8\pi^2\kappa^2} A^2\,. \tag{4.7}$$

Let us first review the quantization of the classical moduli space of LLM in [37, 38]. Consider the geometry with a single black region located around the origin ($x_1 = x_2 = 0$). We adopt the spherical coordinates $(r, \phi)$ and parameterize the boundary of the black region by a function $\rho(\phi)$,

$$r^2 = r_0^2 + \rho(\phi)\,, \tag{4.8}$$

and $r_0$ is chosen such that $A = \pi r_0^2$ and $\rho(\phi)$ is assumed to be a small single-valued function. Under this approximation, the classical moduli space is the function space of $\rho(\phi)$ and is treated as the phase space for geometric quantization. The symplectic form was computed in Type IIB supergravity to be

$$\omega = \kappa^{-1} \oint \oint d\phi d\tilde{\phi}\, \mathrm{sign}\, (\phi - \tilde{\phi})\, \delta\rho(\phi) \wedge \delta\rho(\tilde{\phi})\,, \tag{4.9}$$

from which one finds the Poisson bracket

$$\{\rho(\phi), \rho(\tilde{\phi})\} = 8\pi\kappa^2 \delta'(\phi - \tilde{\phi})\,. \tag{4.10}$$

This system can be quantized by promoting the Poisson bracket to a commutator,

$$i\hbar_{10}\{\cdot, \cdot\} \to [\cdot, \cdot]\,, \tag{4.11}$$

where $\hbar_{10}$ is the ten-dimensional Planck constant. If we consider the mode expansion

$$\rho(\phi) = \sum_{n \in \mathbb{Z}} \alpha_n e^{in\phi}\,, \tag{4.12}$$

then the zero mode is fixed by the condition (4.6),

$$\alpha_0 = 0\,, \tag{4.13}$$

and substituting the mode expansion into (4.10) and (4.11) gives the commutator for the nonzero modes

$$[\alpha_m, \alpha_n] = 4\kappa^2 \hbar_{10} n \delta_{m+n}\,. \tag{4.14}$$

The Hamiltonian (4.7) expressed in terms of these modes is

$$H = \frac{1}{4\kappa^2} \sum_{n=1}^{\infty} \alpha_{-n}\alpha_n + \text{constant} , \tag{4.15}$$

and the partition function is

$$Z(\beta) = \text{Tr}\, e^{-\beta H} = \prod_{n=1}^{\infty} \frac{1}{1 - e^{-\beta \hbar_{10} n}} . \tag{4.16}$$

We see that the geometric quantization of the classical moduli space produces the spectrum of half-BPS operators in the infinite $N$ limit, which agrees with finite $N$ up to $E \leq N$ (the trace relations in the dual CFT kick in at $E > N$). This is due to working in an approximation that the fluctuation $\rho(\phi)$ is small.

The $N$-independent answer obtained above is no different from the quantization of linear solutions to Type IIB supergravity, so the small fluctuation approximation failed to capture the nonlinear information contained in the LLM solutions. To remedy this, we turn to a different quantization procedure that involves no approximation.

As explained, the exact classical moduli space of LLM geometries is the function space of $u(x_i)$. To derive the Poisson bracket, we use the consistency condition introduced in [39], which requires that the Poisson bracket on the moduli space should be such that the Hamilton equation

$$\frac{du(x,t)}{dt} = \{u(x_i,t), H\} \tag{4.17}$$

is compatible with the fact that the LLM geometry is stationary, i.e.

$$\frac{du(x,t)}{dt} = (\text{constant}) \times \frac{du(x,t)}{d\phi} . \tag{4.18}$$

A moment's thought shows that the Poisson bracket takes the form

$$\{A[u], B[u]\} := \alpha \int \{a, b\}\, u(x)\, d^2 x , \quad \{a, b\} := \frac{\partial a}{\partial x_1} \frac{\partial b}{\partial x_2} - \frac{\partial b}{\partial x_1} \frac{\partial a}{\partial x_2} , \tag{4.19}$$

where $A[u]$ and $B[u]$ are functionals given by

$$A[u] = \int a(x) u(x) d^2 x , \quad B[u] = \int b(x) u(x) d^2 x . \tag{4.20}$$

The constant $\alpha$ can be fixed by matching the exact Poisson bracket (4.19) with the Poisson bracket (4.10) for small fluctuations. To do so, consider a black region centered at the origin with radius $r^2 = r_0^2 + \rho(\phi)$. Given a function $f(x)$, we construct a functional

$$F[u] = \int u(x) \frac{1}{r} \frac{\partial f(x)}{\partial r} d^2 x = \oint d\phi \int_0^{r_0^2 + \rho(\phi)} dr^2 u(x) \frac{\partial f(x)}{\partial (r^2)} = \oint f(\rho(\phi), \phi) d\phi \tag{4.21}$$

that is only sensitive to the value of $f(x)$ at the boundary of the black region. Introducing $g(x)$ and $G[u]$ in a similar fashion, we find the Poisson bracket to be

$$
\begin{aligned}
\{F[u], G[u]\} &= 4\alpha \int \left\{ \left| \frac{\partial f}{\partial(r^2)}, \frac{\partial g}{\partial(r^2)} \right| \right\} u \, d^2x \\
&= 4\alpha \oint d\phi \int_0^{r_0^2 + \rho(\phi)} dr^2 \left( \frac{\partial^2 f}{\partial(r^2)^2} \frac{\partial^2 g}{\partial\phi\partial(r^2)} - \frac{\partial^2 g}{\partial(r^2)^2} \frac{\partial^2 f}{\partial\phi\partial(r^2)} \right) \\
&= 4\alpha \oint d\phi \, \frac{\partial f}{\partial\rho} \left( \frac{\partial^2 g}{\partial\phi\partial\rho} + \frac{d\rho}{d\phi} \frac{\partial^2 g}{\partial\rho^2} \right) \\
&= 4\alpha \oint \oint \frac{\partial f(\rho(\phi),\phi)}{\partial\rho} \frac{\partial g(\rho(\tilde\phi),\tilde\phi)}{\partial\rho} \delta'(\phi - \tilde\phi) d\phi d\tilde\phi \,,
\end{aligned}
\tag{4.22}
$$

which matches the Poisson bracket (4.10) when

$$
\alpha = 2\pi\kappa^2 \,.
\tag{4.23}
$$

With the Poisson bracket (4.19) at hand, we can canonically quantize the system as before by promoting the Poisson bracket to a commutator. The details of canonical quantization will be presented in Appendix A. Here, we take a shortcut using deformation quantization. We introduce the Moyal $*$-product,[78]

$$
f * g(x_1, x_2) = e^{\frac{i\hbar}{2}(\partial_{x_1}\partial_{y_2} - \partial_{x_2}\partial_{y_1})} f(x_1, x_2) g(y_1, y_2)\big|_{y_i = x_i} \,,
\tag{4.25}
$$

where $\hbar := \kappa\sqrt{\hbar_{10}}$, and deform the Poisson bracket $\{\cdot, \cdot\}$ to the $*$-commutator. In the small $\hbar$ limit, the $*$-commutator reduces to the Poisson bracket (4.19),

$$
\lim_{\hbar_{10}\to 0} \frac{1}{i\hbar}[f, g]_* = \lim_{\hbar_{10}\to 0} \frac{1}{i\hbar}(f * g - g * f) = \{f, g\} \,.
\tag{4.26}
$$

The $*$-commutator introduces non-commutativity to the $x_1$-$x_2$ plane as $[x_1, x_2]_* = i\hbar$, which gives a minimal area $\Delta A = \Delta x_1 \Delta x_2 \sim 2\pi\hbar$ by the uncertainty principle. As we will see later, the total area of the black region (4.6) is exactly $N$ units of the minimal area.

In terms of the $*$-product, the condition $u(x_i) \in \{0, 1\}$ and the Hamiltonian (4.7) become[9]

$$
u * u = u \,, \quad H = \frac{1}{4\pi\kappa^2} \int (x_1^2 + x_2^2) * u \, d^2x - \hbar_{10} \frac{N^2}{2} \,.
\tag{4.27}
$$

---

[7]The Moyal $*$-product was used in [40] to study the quantization of non-relativistic fermions in one space dimension, a description that is manifest on the CFT side.

[8]The $*$-product admits an equivalent integral form as

$$
f * g(x_1, x_2) = \frac{1}{\pi^2\hbar^2} \int d^2x' d^2x'' \, e^{\frac{2i}{\hbar}(x_1' x_2'' - x_2' x_1'')} f(x_1 + x_1', x_2 + x_2') g(x_1 + x_1'', x_2 + x_2'') \,.
\tag{4.24}
$$

[9]One can show that $(x_1^2 + x_2^2) * u = u * (x_1^2 + x_2^2)$.

The general solution to the equation $u * u = u$ was found in [41] to be

$$u(x) = \sum_{n=0}^{\infty} c_n \phi_n(x), \quad \phi_n(x) = 2(-1)^n e^{-\frac{r^2}{\hbar}} L_n\left(\frac{2r^2}{\hbar}\right), \tag{4.28}$$

where $L_n(x)$ are the Laguerre polynomials, and the coefficients $c_n$ take values in $\{0, 1\}$. The Hamiltonian $H$ and the area $A$ in terms of the modes $c_n$ are

$$H = \hbar_{10} \left[\sum_{n=0}^{\infty}\left(n + \frac{1}{2}\right)c_n - \frac{N^2}{2}\right], \quad A = 2\pi\hbar\sum_{n=0}^{\infty} c_n = 2\pi\hbar N, \tag{4.29}$$

and partition function is

$$Z(\beta) = e^{\beta\hbar_{10}\frac{N^2}{2}} \prod_{n=0}^{\infty}(1 + pe^{-\beta\hbar_{10}(n+\frac{1}{2})})\bigg|_{p^N} = \prod_{n=1}^{N} \frac{1}{1 - e^{-\beta\hbar_{10}n}}, \tag{4.30}$$

which exactly matches the partition function of the half-BPS states in (4.5). Furthermore, a nontrivial consistency check is that the area $A$ follows the same quantization rule $A \in 2\pi\hbar\,\mathbb{Z}_{\geq 0}$ from the deformation quantization via $\sum_{n=0}^{\infty} c_n \in \mathbb{Z}_{\geq 0}$ and the flux quantization via $N \in \mathbb{Z}_{\geq 0}$.

## 4.2 Symmetric product orbifolds and the quantization of LM geometries

We now turn to a second family of holographic CFTs, symmetric product orbifolds. Given a seed theory $\mathcal{T}$, the Hilbert space of $\mathrm{Sym}^N\mathcal{T}$ takes the form [42–44]

$$\mathcal{H}_N = \bigoplus_{[g]} \bigotimes_{n=1}^{\infty} S^{N_n}\mathcal{H}_{(n)}^{\mathbb{Z}_n}, \tag{4.31}$$

where the direct sum is over the twisted sectors labeled by the conjugacy classes $[g]$ of the symmetric group $S_N$, and the (graded) symmetric tensor product $S^N\mathcal{H}$ is

$$S^N\mathcal{H} = \left(\underbrace{\mathcal{H} \otimes \mathcal{H} \otimes \cdots \otimes \mathcal{H}}_{N \text{ times}}\right)^{S_N}. \tag{4.32}$$

Each conjugacy class is a product of single-cycle permutations

$$[g] = (1)^{N_1}(2)^{N_2}\cdots(m)^{N_m}, \tag{4.33}$$

where the integers $N_n$ correspond to the partition $N = N_1 + 2N_2 + \cdots + mN_m$. The Hilbert space $\mathcal{H}_{(n)}^{\mathbb{Z}_n}$ is the $\mathbb{Z}_n$ invariant sector of the Hilbert space $\mathcal{H}_{(n)}$ of the seed $\mathcal{T}$ on a circle of

radius $n$. More explicitly, given an operator of dimension $(h_0, \tilde{h}_0)$ in $\mathcal{T}$, we have an operator of dimension

$$h = \frac{h_0}{n} + \frac{c}{24}\left(n - \frac{1}{n}\right), \quad \tilde{h} = \frac{\tilde{h}_0}{n} + \frac{c}{24}\left(n - \frac{1}{n}\right) \tag{4.34}$$

in $\mathcal{H}_{(n)}$, where $c$ is the central charge of $\mathcal{T}$. The $\mathbb{Z}_n$ acts as a $2\pi$ shift on the circle, whose only effect is to impose the integer spin condition,

$$h - \tilde{h} \in \mathbb{Z}. \tag{4.35}$$

Let us rewrite the Hilbert space $\mathcal{H} := \mathcal{H}_{(1)}^{\mathbb{Z}_1}$ of the single-copy theory $\mathcal{T}$ as a direct sum of the ground state $|1\rangle$ and its orthogonal space

$$\mathcal{H} = |1\rangle \oplus \mathcal{H}'. \tag{4.36}$$

The Hilbert space $\mathcal{H}_N$ can be rewritten as

$$\mathcal{H}_N = \bigoplus_{\substack{N_0, N_1, N_2, N_3, \cdots \\ N = N_0 + N_1 + 2N_2 + 3N_3 + \cdots}} |1\rangle^{\otimes N_0} \otimes S^{N_1}\mathcal{H}' \bigotimes_{n=2}^{\infty} S^{N_n}\mathcal{H}_{(n)}^{\mathbb{Z}_n}. \tag{4.37}$$

The covering vector space is[10]

$$\widetilde{\mathcal{H}} = \left(\bigoplus_{N_1=0}^{\infty} S^{N_1}\mathcal{H}'\right) \bigotimes_{n=2}^{\infty} \left(\bigoplus_{N_n=0}^{\infty} S^{N_n}\mathcal{H}_{(n)}^{\mathbb{Z}_n}\right). \tag{4.39}$$

The quotient map is defined by projecting to the subspace satisfying the condition

$$\sum_{n=1}^{\infty} nN_n \leq N. \tag{4.40}$$

Now, let us focus on the NS-NS sector of the $\mathcal{N} = (4, 4)$ $\mathrm{Sym}^N(T^4)$ and $\mathrm{Sym}^N(K3)$ CFTs. It has been suggested that the spectrum of $\frac{1}{4}$-BPS states is constant except at high codimension loci [28]. To compute this generic $Q$-cohomology, we need to deform the theory away from the free orbifold point at least perturbatively. However, the existing results on the conformal perturbation theory of this deformation are quite limited, only for small charges in the untwisted sector [30] and in the large $N$ limit [45]. Therefore, we focus on the part

---

[10]Note that $\widetilde{\mathcal{H}}$ is different from the Fock space of the second quantized string theory considered in [44], which reads

$$\bigoplus_{N=0}^{\infty} \mathcal{H}_N = \bigotimes_{n \in \mathbb{Z}_{\geq 1}} \bigoplus_{N_n=0}^{\infty} S^{N_n}\mathcal{H}_{(n)}^{\mathbb{Z}_n}. \tag{4.38}$$

of the BPS spectrum that is invariant even at the free point. In particular, we focus on chiral-chiral primary operators ($\frac{1}{2}$-BPS) under the global part $\mathfrak{su}(1,1|2)_L \times \mathfrak{su}(1,1|2)_R$ of the $\mathcal{N} = (4,4)$ superconformal algebra. The chiral-chiral primary partition function is

$$Z_N^{\rm cc} = \prod_{n=0}^{\infty} \prod_{r,s=0}^{2} \frac{1}{(1 - p^{n+1} y^{\frac{n+r}{2}} \bar{y}^{\frac{n+s}{2}})^{(-1)^{r+s} h_{r,s}}} \Bigg|_{p^N} , \tag{4.41}$$

where $y$ and $\bar{y}$ are the fugacities of the $\mathrm{SU}(2)_L \times \mathrm{SU}(2)_R$ R-symmetry, and $h_{r,s}$ are the Hodge numbers of $T^4$ or $K3$.

In the large $N$ limit, the Hilbert space of perturbative BPS states in $\mathrm{AdS}_3 \times \mathrm{S}^3 \times M^4$ for $M^4 = T^4$ or $K3$ is dual to the Fock space of single-cycle chiral-chiral primaries and their descendants ($\frac{1}{4}$-BPS) [46–49]. Conjecture 1 then predicts that this Fock space after imposing the stringy exclusion principle (4.40) constitute the complete set of monotone BPS operators. We leave this check for future work.[11]

At finite $N$, by Conjecture 2, the Hilbert space of chiral-chiral primary operators should be given by quantizing the classical moduli space of half-BPS smooth horizonless geometries, which are the Lunin-Mathur (LM) geometries [50–53]. This quantization [39, 54] reproduces the Hilbert space of the chiral-chiral primary operators, and recovers the partition function (4.41) without the symmetry refinement, i.e. in the specialization $y = \bar{y} = 1$. More generally, the Hilbert space of monotone $\frac{1}{4}$-BPS operators was reproduced in [55, 56] by quantizing the classical moduli space of more general smooth horizonless geometries called superstrata [57–64].

We review the quantization of the LM geometries in the $K3$ case in [39, 54]. The LM geometries are solutions to Type IIB supergravity that are asymptotically $\mathbb{R}^{1,4} \times \mathrm{S}^1 \times M^4$ for $M^4 = T^4$ or $K3$ and preserve 8 supercharges [50–53]. In the "near-horizon" limit, dropping the constant part in the warping factors, the asymptotic geometries become $\mathrm{AdS}_3 \times \mathrm{S}^3 \times M^4$.

The LM geometries are parametrized by a closed non-self-intersecting curve in 24 dimensions described by the function $F_i(s)$ for $i = 1, \cdots, 24$ and $s \in [0,1)$. Fixing the 3-form flux on $\mathrm{S}^3$ and the 7-form flux on $\mathrm{S}^3 \times M^4$ equal to $N_1$ and $N_5$, respectively, the curve satisfies

---

[11]In [30], the lifting of untwisted sector states in the symmetric product orbifold was computed, and it was found that at $(h, \bar{h}, j, \bar{j}) = (1,1,0,2)$, the unlifted supergravity spectrum (at the leading $1/N$ order) was partially lifted at the subleading $1/N$ order. However, in this computation, they ignored the mixing between the untwisted sector and the twist-3 sector (mediated by two insertions of the twist-2 marginal operator), since it does not contribute to the leading $1/N$ order.[12] In light of Conjecture 1, we expect that with this mixing taken into account, no further lifting should occur at all subleading $1/N$ orders, and the lowering of degeneracies as $N$ is decreased is solely an effect of $I_N$, i.e the "stringy exclusion principle". While this is an interesting computation left for future work, one could already argue at this point that the lifting cannot happen because the BPS states with $(h, \bar{h}, j, \bar{j}) = (1,1,0,2)$ at infinite $N$ must be chiral primaries or their descendants.

the constraint

$$\int_0^1 (\dot{F}_i)^2 ds = (2\pi)^2 N_1 N_5 = (2\pi)^2 N \,. \tag{4.42}$$

The symplectic form on the classical moduli space has been computed using supergravity by [39, 54],

$$\omega = \frac{1}{2\pi} \int_0^1 \delta \dot{F}_i \wedge \delta F_i ds \,, \tag{4.43}$$

which gives the Poisson bracket

$$\{F_i(s), \dot{F}_j(\tilde{s})\} = -\pi \left[ \delta(s - \tilde{s}) - 1 \right] \,, \tag{4.44}$$

where on the right-hand side we subtracted off the constant mode, which does not participate in the symplectic form (4.43).

Now, let us quantize the system by promoting the Poisson bracket (4.44) to a commutator

$$[a_{i,m}, a_{j,n}^\dagger] = \hbar \delta_{ij} \delta_{m,n} \,, \tag{4.45}$$

where the oscillator modes are given by expanding $F_i(s)$ as

$$F_i(s) = \sum_{n=1}^\infty \frac{1}{\sqrt{2n}} \left( a_{i,n} e^{2\pi i n s} + a_{i,n}^\dagger e^{-2\pi i n s} \right) \,. \tag{4.46}$$

The constraint (4.42) interpreted as an operator equation as

$$N_1 N_5 = \frac{1}{(2\pi)^2} \int_0^1 : (\dot{F}_i)^2 : ds = \sum_{n=1}^\infty n a_{i,n}^\dagger a_{i,n} \,. \tag{4.47}$$

Hence, the Hilbert space is finite-dimensional, and its dimension is given by

$$\prod_{n=0}^\infty \frac{1}{(1 - p^{n+1})^{24}} \Big|_{p^N} \,, \tag{4.48}$$

which matches exactly with the partition function of the chiral-chiral primary operators (4.41) with $y = \bar{y} = 1$.

# 5 Discussions and open problems

In this work, we introduced a classification of BPS operators in holographic CFTs into monotone and fortuitous. We conjectured that the monotone BPS operators are dual to supersymmetric smooth horizonless geometries, and the fortuitous ones account for the majority of the black hole entropy. Supporting evidence was found in the $\mathcal{N} = 4$ SYM and in

the symmetric product orbifolds $\text{Sym}^N(T^4)$ and $\text{Sym}^N(K3)$. In particular, the quantization of the classical moduli space of the Lin-Lunin-Maldacena geometries produced the *exact* finite-$N$ Hilbert space of half-BPS operators in the $\mathcal{N} = 4$ SYM, and a similar quantization for the Lunin-Mathur geometries and superstrata produced the Hilbert spaces of chiral-chiral primary operators (and more general monotone $\frac{1}{4}$-BPS operators) in the symmetric product orbifolds.

In the $\mathcal{N} = 4$ SYM, the $\frac{1}{4}$-BPS operators in the $\mathfrak{su}(2)$ sector and the $\frac{1}{8}$-BPS operators in the $\mathfrak{su}(2|3)$ sector are dual to the supersymmetric geometries constructed in [32–36]. It would be interesting to see if the quantization of their classical moduli spaces correctly reproduces the Hilbert spaces on the CFT side, and to identify the dual supersymmetric geometries corresponding to more general monotone operators in the $\mathcal{N} = 4$ SYM, such as the $\frac{1}{8}$-BPS operators in the $\mathfrak{su}(1, 1|2)$ and $\mathfrak{su}(1, 2|2)$ sectors. Moreover, since this method does not manifestly rely on supersymmetry and asymptotic AdS, it is tempting to apply this method to more general backgrounds, such as non-supersymmetric asymptotically AdS spaces or even flat and de-sitter spaces, to *derive* more general holographic principles.

What is the bulk dual of a *typical* fortuitous BPS operator?[13] By Conjecture 2 and 3, we expect that the bulk dual cannot be a smooth horizonless geometry, so one guess is that we need to consider D-branes or other solitonic objects in curved supergravity backgrounds and quantize the moduli space of BPS solutions to the coupled bulk-brane equations, e.g. supergravity coupled to super-Yang-Mills. The recent giant graviton expansion formulae [65–76] for the superconformal index of $\mathcal{N} = 4$ SYM may have an interpretation through such quantizations.

From the bulk perspective, the $N$ dependence of the observables naturally organizes into the perturbative $1/N$ terms coming from gravitational loop effects and the nonperturbative $e^{-N^\#}$ terms coming from brane and instanton corrections. Given a dual sequence of CFT correlators with increasing $N$, one may ask how the expansion in $1/N$ and $e^{-N^\#}$ could be extracted. A natural definition of the $1/N$ expansion coefficients is to take the $N \to \infty$ limit of finite-difference approximations to $\partial_N^n$, and perhaps the resurgence of this series could provide a nonperturbative completion. One can also ask whether there is a unique analytic continuation from integer $N$ to (a region of) the complex $N$ plane. Carlson's theorem tells us that this amounts to understanding the $N \to i\infty$ limit, but it is unclear how to make sense of this limit when $N$ is the rank of matrices.[14]

---

[13]While we presented nontrivial evidence that bulk states coming from quantizing a singular geometry must be dual to fortuitous states, some bulk states coming from quantizing smooth horizonless geometries can still be fortuitous. We thank one of the anonymous journal referees for prompting us to clarify this point.

[14]By contrast, Renyi entropies have a unique analytic continuation, because $\rho^n$ for $n \in i\mathbb{R}$ can be bounded to satisfy Carlson's condition; see [77] and related references. We thank Xi Dong for a discussion.

In promoting the holographic covering from the level of Hilbert spaces to that of operator algebras, the challenge is to construct an associative algebra $\widetilde{\mathcal{A}}$ with continuous $N$ dependence in such a way that the null ideals $I_N$ consistently decouple at integer values of $N$. For special classes of local operators, such as the chiral ring whose products do not involve any spacetime coordinate dependence, this should be rather straightforward. Perhaps the next target would be the chiral algebra (super-$W_N$ algebra) of 4d $\mathcal{N} = 4$ SYM [78]. With even less supersymmetry, the holomorphic twist [79–82, 18, 83] renders the minimally supersymmetric subsectors of 4d SCFTs into holomorphic theories on $\mathbb{C}^2$, and one could contemplate the same question there.

# Acknowledgments

We are grateful to Ofer Aharony, Kasia Budzik, Yiming Chen, Matthew Heydeman, Gary T. Horowitz, Shota Komatsu, Ji Hoon Lee, Sameer Murthy, Hirosi Ooguri, Mukund Rangamani, Stephen H. Shenker, and especially Xi Yin for inspiring discussions. CC is partly supported by the National Key R&D Program of China (NO. 2020YFA0713000), and YL is supported in part by DOE grant DE-SC0007870. This research was supported in part by grant NSF PHY-2309135 to the Kavli Institute for Theoretical Physics (KITP). We thank the hospitality of the KITP Program "What is String Theory? Weaving Perspectives Together".

# A    Canonical quantization of the LLM geometries

In this appendix, we canonically quantize the LLM geometries. We view the canonical quantization of a classical system in a general sense as finding a quantum system (operator algebra) whose commutators in the classical limit ($\hbar_{10} \to 0$) reduce to the Poisson bracket of the classical system.

Let us consider a non-relativistic fermion field $\psi(x)$ with the anti-commutators

$$\{\psi^\dagger(x), \psi(y)\} = \hbar^{-1}\delta(x - y), \quad \{\psi(x), \psi(y)\} = 0.$$ (A.1)

The Fourier transform of the fermion field $\psi(x)$ is

$$\widetilde{\psi}(p) = \int dx\, e^{-\frac{i}{\hbar}px}\psi(x),$$ (A.2)

which satisfies the anti-commutator

$$\{\psi^\dagger(x), \widetilde{\psi}(p)\} = \hbar^{-1}e^{-\frac{i}{\hbar}px}, \quad \{\widetilde{\psi}(p), \widetilde{\psi}(q)\} = 0.$$ (A.3)

Let us define the bilocal operator

$$u(x_1, x_2) = \hbar e^{\frac{i}{\hbar} x_1 x_2} \psi^\dagger(x_1) \widetilde{\psi}(x_2) \,. \tag{A.4}$$

Using the commutators (A.3), we find that $u(x_1, x_2)$ satisfies the identity

$$u(x_1, x_2)^2 = u(x_1, x_2) \,. \tag{A.5}$$

Hence, $u(x_1, x_2)$ has eigenvalues 0 or 1 in agreement with the values of $u(x_1, x_2)$ in the classical LLM geometries. Next, let us compute the commutator of the functionals $A[u]$ and $B[u]$ given in (4.20),

$$
\begin{aligned}
[A[u], B[u]] &= \int d^2 x d^2 \tilde{x} \, a(x) b(\tilde{x}) [u(x), u(\tilde{x})] \\
&= \int d^2 x d^2 \tilde{x} \, a(x) b(\tilde{x}) e^{\frac{i}{\hbar}(x_1 - \tilde{x}_1)(x_2 - \tilde{x}_2)} \left( u(x_1, \tilde{x}_2) - u(\tilde{x}_1, x_2) \right) \\
&= 2\pi\hbar \int d^2 x \, e^{-i\hbar \partial_{\tilde{x}_1} \partial_{\tilde{x}_2}} \left[ a(x_1, \tilde{x}_2) b(\tilde{x}_1, x_2) - a(\tilde{x}_1, x_2) b(x_1, \tilde{x}_2) \right] u(x_1, x_2) \Big|_{\tilde{x} \to x} \,,
\end{aligned}
\tag{A.6}
$$

where we have used the relation

$$e^{-i\hbar \partial_{x_1} \partial_{x_2}} f(x_1) g(x_2) = \frac{1}{2\pi\hbar} \int d^2 x' \, e^{-\frac{i}{\hbar}(x_1 - x_1')(x_2 - x_2')} f(x_1') g(x_2') \,. \tag{A.7}$$

The Planck constant $\hbar$ of the non-relativistic fermion system is related to the Planck constant $\hbar_{10}$ of the ten-dimensional type IIB supergravity by

$$\hbar = \kappa \sqrt{\hbar_{10}} \,. \tag{A.8}$$

With this identification, we match the Poisson bracket (4.19) on the classical moduli space of the LLM geometries with the $\hbar_{10} \to 0$ limit of the commutator as

$$\lim_{\hbar_{10} \to 0} \frac{1}{i\hbar_{10}} [A[u], B[u]] = 2\pi\kappa^2 \int d^2 x \left( \frac{\partial a}{\partial x_1} \frac{\partial b}{\partial x_2} - \frac{\partial a}{\partial x_2} \frac{\partial b}{\partial x_1} \right) u = \{A[u], B[u]\} \,. \tag{A.9}$$

Substituting (A.4) into the area (4.6) and the Hamiltonian (4.7), we find

$$
\begin{aligned}
A &= 2\pi\hbar^2 \int \psi^\dagger(x)\psi(x) dx = 2\pi\hbar N \,, \\
H &= \frac{\hbar}{4\pi\kappa^2} \int dx dp \, (p^2 + x^2) \, e^{\frac{i}{\hbar} px} \psi^\dagger(x) \widetilde{\psi}(p) - \hbar_{10} \frac{N^2}{2} \\
&= \hbar_{10} \left[ \int dx \, \psi^\dagger(x) \left( -\frac{\hbar^2}{2} \frac{d^2}{dx^2} + \frac{1}{2} x^2 \right) \psi(x) - \frac{N^2}{2} \right] \,.
\end{aligned}
\tag{A.10}
$$

Let us consider the mode expansion of the fermion field $\psi(x)$ as

$$\psi(x) = \hbar^{-\frac{1}{2}} \sum_{n=0}^{\infty} c_n \Psi_n(x) \,, \tag{A.11}$$

where $\Psi_n(x)$ are the normalized eigenfunctions

$$\left( -\frac{\hbar^2}{2} \frac{d^2}{dx^2} + \frac{1}{2} x^2 \right) \Psi_n(x) = \hbar \left( n + \frac{1}{2} \right) \Psi_n(x) \,, \quad \int_{-\infty}^{\infty} |\Psi_n(x)|^2 d^2 x = 1 \,. \tag{A.12}$$

In terms of the modes $c_n$'s, we find

$$N = \sum_{n=0}^{\infty} c_n^\dagger c_n \,, \quad H = \hbar_{10} \left[ \sum_{n=0}^{\infty} \left( n + \frac{1}{2} \right) c_n^\dagger c_n - \frac{N^2}{2} \right] \,, \tag{A.13}$$

where the first equation identifying the 5-form flux number $N$ with the fermion number gives a nontrivial consistency check of our quantization. Let $|0\rangle$ be the vacuum state annihilated by the annihilation operators $c_n$. The excitation states are given by acting the creation operator $c_n^\dagger$ on $|0\rangle$. The first equation in (A.13) fixes the total fermion number equals $N$. Therefore, the Hilbert space of the system is spanned by the states

$$c_{n_1}^\dagger \cdots c_{n_N}^\dagger |0\rangle \,. \tag{A.14}$$

The partition function is

$$Z(\beta) = \operatorname{Tr} e^{-\beta H} = \sum_{\substack{n_i \in \mathbb{Z}_{\geq 0} \\ n_1 < n_2 < \cdots < n_N}} e^{-\beta E(n_i)} = \prod_{n=1}^{N} \frac{1}{1 - e^{-\beta \hbar_{10} n}} \,,$$

$$E(n_i) = \hbar_{10} \left[ \sum_{i=1}^{N} (n_i + \frac{1}{2}) - \frac{N^2}{2} \right] \,, \tag{A.15}$$

which is equal to the partition function (4.30) computed using the deformation quantization.

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
