# Peer review of "Holographic covering and the fortuity of black holes"

_SciPost Physics_

## Round 1 · Referee Report · Anonymous (Referee 1) · 2025-12-21

Report

The authors propose a classification of BPS states in a sequence of CFTs labelled by rank N based on the behavior of the supercharge cohomology classes with respect to N.

The work provides a conceptual and mathematical framework for the understanding of BPS states at finite N that do not admit a lift to the covering Hilbert space, based on the structure of trace relations in a sequence of CFTs labelled by N. The findings represent a novel approach to the study of BPS states in holographic CFTs and have very interesting implications for the study of supersymmetric black holes.

We should, however, point out that we find the arguments for Conjecture 2 and its supporting evidence in Section 4 weaker compared to the rest of the paper:

  1. Do the authors intend to say that all monotone BPS states at finite N can be found by quantizing the classical moduli space of supersymmetric, smooth and horizonless solutions? The evidence for this statement in the existing literature is weak for less supersymmetric geometries. As the authors mention, the quantization of the moduli space of geometries in less supersymmetric sectors has not been carried out (even at large N) in AdS5 x S5. Furthermore, it is stated on page 18 that “the Hilbert space of monotone 1/4-BPS operators was reproduced in [55,56] by quantizing the classical moduli space of more general smooth horizonless geometries called superstrata.” However, it appears to us that the analysis done in [55,56] captures only a fraction of monotone 1/4-BPS states that are located below the de Boer bound.

  2. Based on the examples presented, verification of Conjecture 2 in specific contexts is based on generating functions that contain the finite N partition function as the coefficient of p^N. However, these generating functions are also known to exist in contexts where BPS black holes are present, such as in the (modified) elliptic genus of symmetric orbifolds. As these generating functions are Fock space partition functions, it is not clear that they cannot also be reproduced by the quantization of an intricate moduli space. Could the authors clarify what structural features distinguish the generating functions for monotone versus fortuitous states? Specifically, what would prevent the quantization of a sufficiently intricate moduli space from reproducing fortuitous (index) generating functions, given that they both admit Fock space representations?

The potential for sharpening Conjecture 2 does not diminish the paper's overall contribution. I recommend publication in SciPost Physics once the authors have responded to the points raised above.

Recommendation

Ask for minor revision

---

## Editorial Decision

in_refereeing